# The Usefulness of Vitamin K-Dependent Proteins in the Diagnosis of Colorectal Carcinoma

**DOI:** 10.3390/ijms25094997

**Published:** 2024-05-03

**Authors:** Mirela-Georgiana Perné, Adela-Viviana Sitar-Tăut, Olga Hilda Orășan, Vasile Negrean, Călin Vasile Vlad, Teodora-Gabriela Alexescu, Mircea Vasile Milaciu, Lorena Ciumărnean, Răzvan Dan Togănel, Gabriel Emil Petre, Ioan Șimon, Alexandra Crăciun

**Affiliations:** 14th Department–Internal Medicine, 4th Medical Discipline, Faculty of Medicine, “Iuliu Haţieganu” University of Medicine and Pharmacy, Republicii Street, Nr. 18, 400015 Cluj-Napoca, Romania; 26th Department–Surgery, 4th Surgery Discipline, Faculty of Medicine, “Iuliu Haţieganu” University of Medicine and Pharmacy, Republicii Street, Nr. 18, 400015 Cluj-Napoca, Romania; 32nd Department–Molecular Sciences, Discipline of Medical Biochemistry, Faculty of Medicine, “Iuliu Haţieganu” University of Medicine and Pharmacy, Pasteur Street, Nr. 6, 400349 Cluj-Napoca, Romania

**Keywords:** vitamin K, PIVKA II, ucMGP, CEA, CA19-9, colorectal cancer, diagnosis

## Abstract

Colorectal cancer (CRC) is one of the most common neoplasms in developed countries, with increasing incidence and mortality, even in young people. A variety of serum markers have been associated with CRC (CEA, CA 19-9), but neither should be used as a screening tool for the diagnosis or evolution staging of CRC. The sensitivity and specificity of these markers are not as good as is required, so new ones need to be found. Matrix Gla protein and PIVKA II are involved in carcinogenesis, but few studies have evaluated their usefulness in predicting the presence and severity of CRC. Two hundred patients were divided into three groups: 80 patients were included in the control group; 80 with CRC and without hepatic metastasis were included in Group 1; 40 patients with CRC and hepatic metastasis were included in Group 2. Vitamin K-dependent proteins (VKDPs) levels in plasma were determined. Patients with CRC without methastasis (Group 1) and CRC patients with methastasis (Group 2) presented significantly higher values of CEA, CA 19-9, PIVKA II (310.05 ± 38.22 vs. 430.13 ± 122.13 vs. 20.23 ± 10.90), and ucMGP (14,300.00 ± 2387.02 vs. 13,410.52 ± 2243.16 vs. 1780.31 ± 864.70) compared to control group (Group 0). Interestingly, Group 1 presented the greatest PIVKA II values. Out of all the markers, significant differences between the histological subgroups were found only for ucMGP, but only in non-metastatic CRC. Studying the discrimination capacity between the patients with CRC vs. those without, no significant differences were found between the classical tumor markers and the VKDP AUROC curves (PIVKA II and ucMGP AUROCs = 1). For the metastatic stage, the sensitivity and specificity of the VKDPs were lower in comparison with those of CA 19-9 and CEA, respectively (PIVKA II AUROC = 0.789, ucMGP AUROC = 0.608). The serum levels of these VKDPs are significantly altered in patients with colorectal carcinoma; it is possible to find additional value of these in the early stages of the disease.

## 1. Introduction

Colorectal cancer (CRC) is one of the most common neoplasms in developed countries [1,2,3], representing a public health problem. CRC incidence and mortality vary significantly around the world [4]. CRC is the third most frequently diagnosed cancer in men and the second in women worldwide, according to the World Health Organization’s GLOBOCAN database [4,5].

Approximately 153,000 new cases of CRC are diagnosed annually, of which 107,000 cases are colon cancer and over 46,000 cases are rectal cancer [1,6]. In the United States, the incidence of CRC decreased by about 2% per year between 2014 and 2018 [6], in contrast to other Western countries, where the incidence remained relatively constant or even increased [3]. The colorectal carcinoma incidence has increased in historically low-risk areas, such as Spain, East Asia, and Eastern Europe [7], being (the same as mortality) substantially higher in men than in women [8]. Theoretically, CRC is uncommon before the age of 40, but it should be noted that recent studies have shown an alarming increase in the incidence in the 40–49 years age group [6,9,10,11,12,13,14,15,16], which has recently led to a reduction in the recommended screening age (at 40 years for patients with a family history of colorectal neoplasm) [17]. Environmental factors and genetic factors lead to the risk of developing CRC [7].

The positive diagnosis of CRC is established by histological examination of a biopsy obtained during lower digestive endoscopy or from a surgical sample [18]. Histopathological, most cancers that occur in the colon and rectum are adenocarcinomas [19,20]. CT colonography provides an equally sensitive and less invasive diagnostic alternative compared to colonoscopy for patients who have symptoms suggestive of CRC, but does not allow removal of the tumor lesion or biopsy sampling [21,22].

Regarding laboratory paraclinical examinations, a variety of serum markers have been associated with CRC, especially carcinoembryonic antigen (CEA), currently the most used in colorectal cancer [23]. However, studies have shown that these have a low diagnostic capacity in the detection of primary CRC in its early stages [23,24,25]. According to the recent meta-analysis published in 2018 by Liu [1], the CEA sensitivity for CRC diagnosis was 46% (95% CI 0.45–0.47) and its specificity was 89% (95% CI 0.88–0.92). High serum levels of CEA can be found also in non-oncological pathologies (gastritis, peptic ulcer, diverticulitis, liver disease, pancreatitis, renal failure, diabetes, in any acute or chronic inflammatory state) [26]. Furthermore, the serum levels of CEA are significantly higher in smokers than in non-smokers [27,28]. no other conventional tumor marker had a better diagnostic sensitivity. Carbohydrate antigen 19-9 (CA 19-9) exhibiting an even lower sensitivity, of about 30% (95% CI 0.28-0.32), with a specificity of 92% (95% CI 0.915–0.940) for CRC detection [1].

Thus, according to the literature [29], neither serum CEA nor CA 19-9 should be used as a screening test for the diagnosis or staging of CRC [30]. However, what is clearly established is the usefulness of these markers in tracking patients with CRC, and in establishing the medium- and long-term prognosis [23].

### Vitamin K and Extrahepatic Vitamin K-Dependent Proteins

Early works in this area pointed out that vitamin K is a cofactor of gamma-glutamyl carboxylase, which catalyzes the conversion of glutamate residues to carboxy glutamate (Gla) [31]; these vitamin K-dependent proteins are known as Gla proteins [32]. They play an important role in blood coagulation, bone metabolism, and carcinogenesis.

The role of vitamin K at the intestine level has raised considerable attention in recent years. Published studies have proposed the idea that an insufficient intake of vitamin K in the diet is associated with an increase in the incidence, metastasis appearance, and mortality of oncological diseases (colorectal, hepatocarcinoma, prostate cancer) [33,34,35,36].

Vitamin K-dependent Proteins—VKDPs—have been named Gla proteins, and more than 16 of these proteins have been identified [32,37]. For a long time, research has focused only on Gla liver proteins involved in coagulation [38]. Recently, new data have revealed the important involvement of vitamin K-dependent proteins in various extrahepatic functions, such as arterial calcifications, atherosclerosis, bone metabolism, inflammation, and carcinogenesis [3,32,37,38,39,40,41,42].

In 2017, the first review in extenso, written by Dahlberg et al. [43] (in relation to the possible association between the serum levels of vitamin K-dependent proteins and colorectal carcinoma), highlighted the importance of determining subclinical vitamin K deficiency and vitamin K-dependent proteins in association with neoplasia [43].

PIKVA-II (Protein Induced by Vitamin K Absence—II) was first presented in 1984 [36]. In the absence of vitamin K or when its action is antagonized (for example, by oral anticoagulants like warfarin), PIVKA-II is released into the blood. PIVKA-II is used to estimate the hepatic status of vitamin K and is more sensitive for detecting vitamin K deficiency than standard clotting tests, such as those in [43,44].

The expression of PIVKA II in different oncological pathologies was evaluated by previous research. PIVKA-II levels reflect oncogenesis and progression of HCC [40] with a better diagnostic accuracy in diagnostic HCC vs. alpha-fetoprotein [45]. Dong, Shirabe, Inagaki, Yu [3,46,47,48] showed that in patients with gastrointestinal neoplasms, PIVKA-II levels were above the reference range in most patients, speculating that PIVKA II could play an important role in tumor angiogenesis, in microvascular invasion of cancer cells.

Matrix Gla Protein (MGP) is synthesized by chondrocytes or endothelial cells. Initially, MGP was thought to have the unique role of binding calcium ions and inhibiting calcification [2,32,49]. It was demonstrated that the inactive form of MGP can be considered a marker for the assessment of vascular calcification [50]. Additionally, MGP is involved in a variety of cancers trough the overexpression of the MGP gene in breast cancer cells, in primary renal, testicular, and prostate carcinomas, in ovarian and digestive neoplasia, and in glioblastoma [2,3,33,38,49,50,51,52,53].

In addition, it appears to have a role in tumor angiogenesis; a direct relationship between the expression of MGP and tumor vascularization has been reported [38,49].

The purpose of this study was to extend our knowledge on the benefit of using vitamin K-dependent proteins in the diagnosis and follow-up of patients with colorectal carcinoma, to study whether vitamin K-dependent proteins are useful non-invasive markers in predicting the presence and severity of CRC.

The analysis of the correlations between the serum levels of PIVKA-II, ucMGP, CEA, and CA 19-9 was performed in patients with colorectal carcinoma with and without liver metastases in comparison with a control group. The serum levels of PIVKA II, ucMGP, CEA, and CA 19-9 were assessed in order to evaluate the possibility of their use in the screening, diagnosis, and follow-up of colorectal cancer.

## 2. Results

The demographic and clinical data of the patients studied are presented in Table 1. There were no significant differences between the three groups regarding the average age or the distribution of patients according to their gender, respectively. Symptoms were present only in patients with carcinomas; however, there were no significant differences between the two groups (Groups 1 and 2).

The most frequent location of carcinoma was at the level of the sigma and rectum, but without significant differences between Group 1 and 2 (Table 2), and from a histological point of view, the G2 and G3 tumoral differentiated types were more frequent in these groups.

Significant differences were found between the groups regarding the CEA, CA 19-9, PIVKA II, and ucMGP values. Group 1 and Group 2 presented significantly higher values not only of the tumor markers but also of ucMGP and PIVKA-II compared to Group 0. Interestingly, the CEA, CA 19-9, and ucMGP values were greater in Group 2 vs. Group 1; but Group 1 presented greater PIVKA II values in comparison with the Group 2 values.

As it is shown in Table 3, considering the histological type, in the G2 and G3 types, Group 2 presented greater CEA, CA 19-9, and ucMGP values in comparison with Group 1. The PIVKA II values registered in the G2 and G3 subgroups were greater in the patients without metastasis vs. those with metastasis.

Out of all the markers, significant differences between the histological subgroups were registered only for ucMGP, but only in those with non-metastatic CRC.

Overall, in the CRC cases with no metastasis, significant correlations were found between CEA and PIVKA II (inverse correlation) and ucMGP; an inverse correlation was also found between these two vitamin K-dependent proteins’ values (Figure 1 and Figure 2). This relationship was not validated in the metastatic patients. Going further with the analysis, the relationship was partially confirmed in the G2 and G3 types.

Studying the discrimination capacity between the patients with CRC cancer vs. those without CRC, no significant differences were detected between the classical tumor markers and PIVKA II and ucMGP in the AUROC curves. For the metastatic stage, the sensitivity and specificity of the vitamin K-dependent proteins were lower in comparison with CA 19-9 and CEA, respectively (the complete data are presented in Table 4).

The significant differences between the AUROCs were as follows: the * CEA–PIVKA II difference between areas was 0.211 (95% CI 0.131 to 0.291, p < 0.001); the * CEA–ucMGP difference between areas was 0.392 (95% CI 0.282 to 0.501, *p* < 0.001); the * CA 19-9–PIVKA II difference between areas was 0.211 (95% CI 0.131 to 0.291, *p* < 0.001); the * CA 19-9–ucMGP difference between areas was 0.392 (95% CI 0.282 to 0.501, *p* < 0.001); the * PIVKA II–ucMGP difference between areas was 0.180 (95% CI 0.131 to 0.291, *p* < 0.001).

## 3. Discussion

The increasing incidence of CRC, both in young and elderly patients, and the difficulty of obtaining an early and accurate diagnosis, along with the percentage of cases in which the diagnosis is late, have raised the need for detecting new diagnostic tools [6,25].

CRC screening is based on three categories of tests, consisting of the determination of occult bleeding in the stool and endoscopic and imaging evaluation [1,4]. The lower digestive endoscopy provides the basis of the diagnosis, but recent data recommend the additional use of imaging methods (abdominal ultrasound, CT, PET/CT, or MRI) [54] to improve the diagnostic accuracy. The use of CEA and CA 19-9 tumor markers alone is not recommended for screening [1], but they are useful for following the evolution of CRC.

However, studies have shown that combining lower digestive endoscopy with imaging methods and the determination of tumor markers can provide added value in the diagnosis of difficult cases [18]. Currently, the most used markers are represented by CEA and CA 19-9; their specificity and sensitivity determine the existence of the limits of their use [29]. The studies conducted so far have clearly shown that the serum value of these markers is influenced by various pathological conditions besides CRC [26].

Considering the need to identify as early as possible both the occurrence of gastrointestinal neoplasia and the presence of microvascular invasion and liver metastasis appearance, in the context of the absence of “perfect” tumor markers, it is necessary to carry out studies on possible alternative mechanisms involved in tumoral genesis and studies to identify new markers with practical utility. At the same time, the discovery of new ways to promote tumoral genesis provides an opportunity for more effective colorectal cancer screening and allows for the identification of the moment of the evolution of neoplasia.

The discovery and use of new markers is beneficial in terms of user-friendliness [1]. The existing limits, mainly related to theoretical knowledge, with conflicting published data [1], emphasize the necessity to address studies for new tumor serum marker identification (to be used independently or in association with classical ones) with usefulness in the screening and early diagnosis of colorectal cancer.

The mechanisms of vitamin K at the intestinal level consist of an antioxidant effect; it reduces oxidative injury and affects the redox homeostasis of cells, acting like a “radical repair agent”; it causes immunomodulator and anti-inflammatory effects, which inhibits T cell proliferation and NF-kB activation, reducing the IL-6 level [31,34,36]. Vitamin K is also produced by the intestinal microbiota, suppressing gut risk microbes, improving intestinal bacteria flora and promoting beneficial microbial metabolites [31]. In addition, vitamin K possesses antitumor action against various neoplastic cell lines [31,40] by suppressing cyclin D1 expression, inhibiting NF-κB activation, or inhibiting protein kinase C (PKC)-alpha and epsilon kinase activities [2,40], xenobiotic receptor inactivation [33], oxidative stress regulation [33], and also by preventing mitochondrial dysfunction caused by the vitamin K binding Bcl-2 antagonist killer [31,34]. Vitamin K inhibits cancer cell growth [34]. Other interesting papers [40] have shown that vitamin K promotes tumor apoptosis, inducing cytotoxicity in cancer cells [34,35,36].

There are theories, according to which, a subclinical vitamin K deficiency (demonstrated by determining the matrix protein Gla and PIVKA-II levels) could be a potential marker for identifying the occurrence, development, microvascular invasion, and hepatic metastasis in colorectal carcinoma [2,3,48].

Vitamin K-dependent proteins (ucMGP and PIVKA II) represent a class of proteins that initially hold only roles unrelated to the emergence and progression of tumor formation. Recent studies have demonstrated their involvement in tumor angiogenesis and in the occurrence of metastases, leading to the idea that these vitamin K-dependent proteins could be applied not only for neoplasm diagnosis but also for establishing the disease’s evolution timing. This topic has attracted considerable attention, but only a few studies have explored the usefulness of vitamin K-dependent proteins in the diagnosis of CRC, their role being, at this moment, incompletely elucidated. The area of vitamin K-dependent proteins’ involvement in colorectal carcinoma genesis or diagnosis needs further study [33,40].

The exact role of ucMGP in cell differentiation and tumor genesis has not yet been completely clarified [2,50]. The mechanisms by which MGP promotes carcinogenesis are extremely varied, being described as a possible aspect of JAK2/STAT5, TGFβ, NFκB, and intracellular calcium signaling, promoting calcium influx, upregulating PD-L1 expression, and promoting CD8 T cell exhaustion [2,3,33,53].

Caiado H et al. [49] reported, in 2018, the first study showing that increased levels of mRNA expression of MGP are associated with a worse prognosis of CRC, with the occurrence of metastases [3] associated with a lower chance of survival [49]; these results were confirmed in the following years by others [53]. Huang et al. [53] also showed that MGP suppression inhibits tumor proliferation and studied MGP’s role in oncologic treatment. He suggested the possible reversal of oxaliplatin resistance in CRC [53] by upregulating copper transporter 1 and downregulating ATPase copper transporting alpha (ATP7A) and ATPase copper transporting beta (ATP7B) [53]. In 2022, Rong D et al. published data showing that the inhibition of MGP reduced liver metastasis and, moreover, increased the efficiency of αPD1 treatment in CRC [3].

The present study reinforced the idea that the serum levels of vitamin K-dependent proteins (PIVKA II and ucMGP) are elevated in oncological patients. In our study, the registered values were much higher in the patients with CRC, with or without metastasis, in comparison with the patients without oncological pathology.

If the CEA, CA 19-9, and ucMGP values increased with disease evolution, and if the PIVKA II values decreased, an inverse U shape for PIVKA II’s evolution with respect to CRC spread would occur (430.13 ± 122.13, median value 451.75 vs. 310.05 ± 38.22, median value 316.70, *p* < 0.0001). At the same time, these data were not dependent on the histological type (G2 or G3, *p* < 0.05 for both PIVKA II and ucMGP).

To the best of our knowledge, a single article has been published so far regarding the relationship between the presence of colorectal neoplasms with secondary dissemination in the liver and the serum level of PIVKA-II [55]. Our results are contrary to those reported by Kato [56]; he reported a high value of PIVKA II in a patient with CRC cancer and metastasis. This is the only observational published study; the other research is focused on theoretical aspects of VKDPs’ involvement in CRC.

A significant negative correlation was identified between CEA and PIVKA II and a positive correlation was identified between CEA and ucMGP in patients without metastasis. The results reveal that a correlation is not present in metastatic disease, neither for ucMGP, nor for PIVKA II. A possible explanation for our results is that vitamin K-dependent proteins are synthesized in the liver, the moment of metastasis being associated with the decrease depending on the altered synthesis of the affected liver (previous studies suggest that even jaundice is associated with decreased values of PIVKA II [56]). Vitamin K, through the action of inducing apoptosis, is involved in inhibiting cancer metastasis [34]. A vitamin K deficit revealed by increased values of ucMGP and PIVKA-II could possibly be a potential marker in the early evolutionary stages of CRC. However, we do not have a clear explanation at this moment about the opposite relation between CEA and PIVKA II and ucMGP, respectively.

No significant differences have been registered regarding the ability of tumor or metastasis diagnosis. Even though the sensitivity for PIVKA II and ucMGP was good, the specificity was lower in comparison with CEA and CA 19-9. Microvascular invasion cannot be detected by imagistic evaluation, so it is possible for some of the cases (included in the group theoretically without metastasis) to already present microvascular invasion. The idea is supported by the significant difference in values registered between case-control subjects and CRC patients. Taking into consideration this idea, the next research must focus on the relationship between ucMGP and PIVKA-II values regarding patient evolutive stages.

To the best of our understanding, no previous study has focused on vascular and hepatic uncarboxylated vitamin K-dependent proteins in CRC.

Our study is the first that has aimed to compare commonly used tumoral markers with uncarboxylated VKDPs. As we have mentioned previously, this is just a pilot study, so further research is necessary for a detailed analysis of the role of ucMGP and PIVKA-II in colorectal cancer.

The study’s limitations include the relatively small number of patients; further exhaustive research is needed. Another important limitation of the study is the fact that some possible confounders must be evaluated, with the relationship with other pathologies (vascular or osteoarticular diseases) or physiological situations not being established. Also, our study did not collect data about the associated diseases.

However, the main limitation of our study is the absence of a correlation with an anatomopathological evaluation, with the complete data about microvascular invasion not being available.

## 4. Materials and Methods

This was an analytical, prospective, observational, and case–control study, which included a total of 200 patients, divided into three groups—80 patients without oncological pathology (Group 0), 80 patients with colorectal carcinoma without secondary liver dissemination (Group 1), and 40 patients with colorectal cancer and secondary liver dissemination (Group 2).

Subjects were selected from patients who presented at the Internal Medicine and Gastroenterology departments within the 4th Medical Clinic, CF Clinical Hospital, Cluj-Napoca, and who were hospitalized for multidisciplinary evaluation. For each subject, demographic data (age, gender, comorbidities, symptoms—rectal bleeding, constipation, weight loss), and the results of clinical and paraclinical examinations were registered.

Patients diagnosed with colorectal carcinoma, with (Group 2) and without (Group 1), secondary liver determinations, were included in the study. The diagnosis was established based on the clinical examination and lower digestive endoscopy with biopsies of the tumor formation and positive histopathological examination for colorectal adenocarcinoma, considering the histopathological classification of the degree of tumor differentiation (G1 = well-differentiated (low-grade) tumor; G2 = moderately differentiated (intermediate-grade) tumor; G3 = poorly differentiated (high-grade) tumor; G4 = undifferentiated (high-grade) tumor). For all subjects included in this study, abdominal–pelvic CT with contrast agent was performed to localize and assess the stage of the disease using the colorectal cancer TNM classification.

For the control group, disease-free subjects were chosen, proven by negative lower digestive endoscopy for the presence of polyps or tumor formations.

Patients who refused to participate in this study, and patients presenting other comorbidities (other neoplasia, inflammatory chronic disease—intestinal, systemic erythematosus lupus, scleroderma–osteoporosis, ischemic heart diseases, stroke, liver cirrhosis regardless of etiology) or interfering treatments (vitamin K or antivitamin K treatment).

For each patient, serum levels of ucMGP, PIVKA-II, CEA, and CA 19-9 were evaluated.

Determination of ucMGP, PIVKA-II: After taking the biological samples, they were centrifuged and stored at −80 °C until the examination. Subsequently, the samples were processed from plasma by the ELISA method, using kits produced by CUSABIO. This method involves immobilizing the antibody on a solid surface, to which the serum to be investigated is added. If the serum contains the specific antigen, it will bind to the immobilized antibody. The concentration of the antigen in the serum was determined spectrophotometrically.

Statistical analysis: The statistical analysis was carried out using the statistical packages Medcalc version 10.3.0.0 (MedCalc Software, Ostend, Belgium). The Kolmogorov–Smirnov and D’Agostino–Pearson tests were used for normality distribution assessment; numerical data were presented as mean ± standard deviation or median values; qualitative data as numbers and percentages. The differences between data were evaluated with the independent sample *t*-test, Mann–Whitney test, Chi 2-test, ANOVA, Kruskal–Wallis test. Spearman’s and Pearson’s correlation coefficients were calculated. Area under receiver operating characteristic (ROC) curves (AUROC) was calculated. A *p*-value < 0.05 was considered statistically significant.

## 5. Conclusions

The results of this study suggest that the serum levels of ucMGP and PIVKA-II are significantly altered in patients with colorectal carcinoma; we assume this occurs in the early stages of the disease. Additional studies are necessary to establish the predictive values of ucMGP and PIVKA-II proteins in CRC.

## Figures and Tables

**Figure 1 ijms-25-04997-f001:**
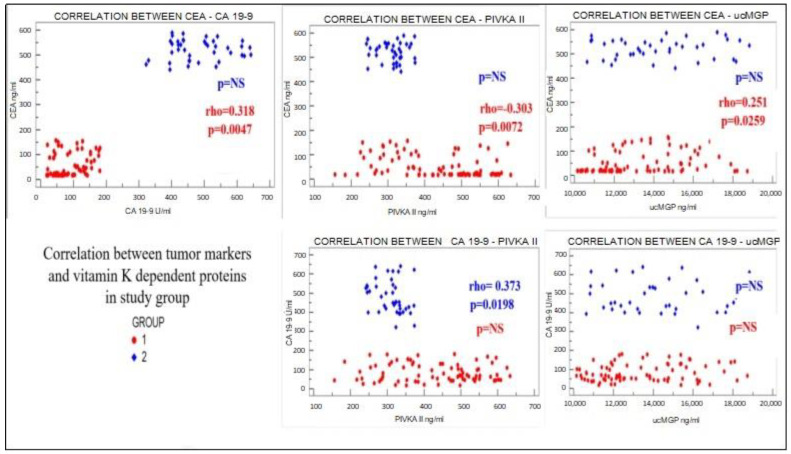
Correlation between tumor markers MGP and PIVKA II in the study group. CEA = carcinoembryonic antigen; CA 19-9 = cancer antigen 19-9; PIVKA II = des-gamma-carboxy prothrombin; ucMGP = uncarboxylated matrix Gla protein.

**Figure 2 ijms-25-04997-f002:**
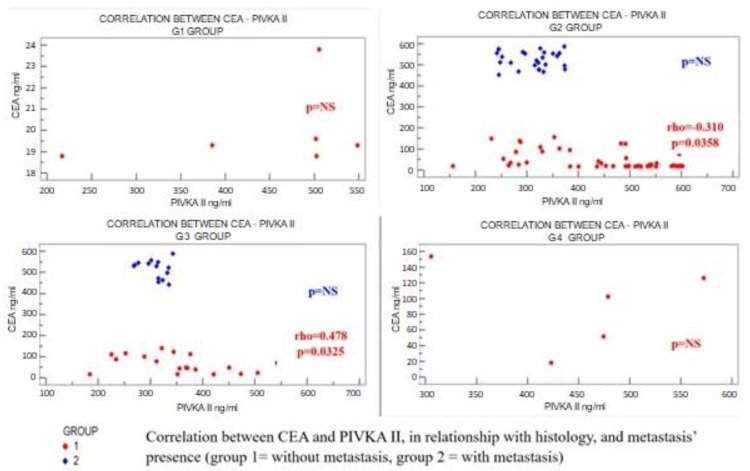
Correlation between CEA and PIVKA II in relation to histology, and metastasis presence. CEA = carcinoembryonic antigen; PIVKA II = des-gamma-carboxy prothrombin; G1 = well-differentiated (low-grade) tumor; G2 = moderately differentiated (intermediate-grade) tumor; G3 = poorly differentiated (high-grade) tumor; G4 = undifferentiated (high-grade) tumor.

**Table 1 ijms-25-04997-t001:** Demographic and clinical characteristics of the patients with CRC.

		Global	Control Group	CRC without Metastasis (Group 1)	CRC with Metastasis (Group 2)	*p*
		No (%)	No (%)	No (%)	No (%)	
Number		200	80	80	40	
Gender	F	69 (34.5%)	33 (41.25)	23(28.75)	13 (32.5)	*p* = 0.2400
	M	131 (65.5%)	47 (58.75)	57 (71.25)	27 (67.5)	
Age *		55.36 ± 6.02 (56)	55.76 ± 5.56 (56)	55.27 ± 6.36 (56)	54.75 ± 6.31 (57)	*p* = 0.7432
Blood in stools	YES	62 (31.0%)	0 (0)	44 (55)	18 (45)	***p*** **< 0.0001**
	NO	138 (69.0%)	80 (100)	36 (45)	22 (55)	
Losing weight	YES	24 (12.0%)	0 (0)	15 (18.75)	9 (22.5)	***p*** **= 0.0001**
	NO	176 (88.0%)	80 (100)	65 (81.25)	31 (77.5)	
Constipation	YES	34 (17.0%)	0 (0)	21 (26.25)	13 (32.5)	***p*** **< 0.0001**
	NO	166 (83.0%)	80 (100)	59 (73.75)	27 (67.5)	

* Does not exhibit normal distribution; CRC = colorectal cancer; the bold values are statistically significant.

**Table 2 ijms-25-04997-t002:** Anatomopathological findings and tumor markers in study groups.

		Global	Control Group (Group 0)	CRC without Metastasis (Group 1)	CRC with Metastasis (Group 2)	*p*-Value
		n (%)	n (%)	n (%)	n (%)	
Localization	Ascending	13 (10.8%)	-	9 (11.25)	4 (10)	*p* = 0.9854
	Transverse	8 (6.7%)	-	6 (7.5)	2 (5)	
	Descending	15 (12.5%)	-	10 (12.5)	5 (12.5)	
	Sigma	24 (20.0%)	-	16 (20)	8 (20)	
	Rectum	60 (50.0%)	-	39 (48.75)	21 (52.5)	

Histology	G1	6 (5.0%)	-	6 (7.5)	0 (0)	*p* = 0.0761
	G2	73 (60.8%)	-	47 (58.75)	26 (65)	
	G3	35 (29.2%)	-	21 (26.25)	14 (35)	
	G4	6 (5.0%)	-	6 (7.5)	0 (0)	
CEA *		126.89 ± 202.55 (18.90)	1.04 ± 0.9 (0.75)	55.04 ± 45.64 (26.50)	522.32 ± 40.78 (530.40)	***p* < 0.0001** significant differences between Group 0 vs. Group 1; Group 0 vs. Group 2; Group 1 vs. Group 2.
CA 19-9		137.85 ± 181.56 (47.65)	19.63 ± 9.94 (19.35)	83.70 ± 45.40 (68.55)	482.57 ± 85.31 (454.90)	***p* < 0.0001** significant differences between Group 0 vs. Group 1; Group 0 vs. Group 2; Group 1 vs. Group 2.
PIVKA II *		242.15 ± 202.94 (277.90)	20.23 ± 10.90 (19.95)	430.13 ± 122.13 (451.75)	310.05 ± 38.22 (316.70)	***p* < 0.0001** significant differences between Group 0 vs. Group 1; Group 0 vs. Group 2; Group 1 vs. Group 2.
ucMGP *		8936.33 ± 6150.42 (11,395.50)	1780.31 ± 864.70 (1653.00)	13410.52 ± 2243.16 (12,780.00)	14300.00 ± 2387.02 (14,031.00)	***p* < 0.0001** significant differences between Group 0 vs. Group 1; Group 0 vs. Group 2; Group 1 vs. Group 2.

* Does not exhibit normal distribution; n = number of subjects, CRC = colorectal cancer; Se = sensibility; Sp = specificity; CEA = carcinoembryonic antigen; CA 19-9 = cancer antigen 19-9; PIVKA II = des-gamma-carboxy prothrombin; ucMGP = uncarboxylated matrix Gla protein; G1 = well-differentiated (low-grade) tumor; G2 = moderately differentiated (intermediate-grade) tumor; G3 = poorly differentiated (high-grade) tumor; G4 = undifferentiated (high-grade) tumor; the bold values are statistically significant.

**Table 3 ijms-25-04997-t003:** CEA, CA 19-9, PIVKA II, ucMGP values in CRC patients, depending on histology and metastasis presence.


**CEA ***		**CRC without Metastasis** **(Group 1)**	**CRC with Metastasis** **(Group 2)**	** *p* **
		Mean ± SD (median)	Mean ± SD (median)	
**Histology**	**G1**	19.93 ± 1.92 (19.30)	-	
	**G2**	53.07 ± 47.00 (23.70)	525.31 ± 39.68 (528.60)	***p*** **< 0.0001**
	**G3**	62.70 ± 41.44 (48.10)	516.77 ± 43.70 (530.40)	***p*** **< 0.0001**
	**G4**	78.68 ± 57.31 (77.30)	-	
** *p* **		*p* = 0.2084	*p* = 0.4784	

**CA 19-9 ***		**CRC without Metastasis** **(Group 1)**	**CRC with Metastasis** **(Group 2)**	** *p* **
		Mean ± SD (median)	Mean ± SD (median)	
**Histology**	**G1**	78.95 ± 21.55 (77.00)	-	
	**G2**	77.15 ± 45.07 (64.10)	485.69 ± 82.48 (468.60)	***p*** **< 0.0001**
	**G3**	94.64 ± 50.16 (97)	476.79 ± 93.24 (452.80)	***p*** **< 0.0001**
	**G4**	101.50 ± 45.03 (103.55)	-	
** *p* **		*p* = 0.4299	*p* = 0.7337	

**PIVKA II ***		**CRC without Metastasis** **(Group 1)**	**CRC with Metastasis** **(Group 2)**	** *p* **
		Mean ± SD (median)	Mean ± SD (median)	
**Histology**	**G1**	443.16 ± 123.57 (501.85)	-	
	**G2**	445.84 ± 122.69 (491.30)	310.23 ± 44.36 (322.35)	***p*** **< 0.0001**
	**G3**	380.96 ± 121.05 (367.80)	309.70 ± 24.45 (314.60)	***p*** **= 0.0251**
	**G4**	466.23 ± 95.13 (476.40)	-	
** *p* **		*p* = 0.1963	*p* = 0.7124	

**ucMGP ***		**CRC without Metastasis** **(Group 1)**	**CRC with Metastasis** **(Group 2)**	** *p* **
		Mean ± SD (median)	Mean ± SD (median)	
**Histology**	**G1**	10,351.83 ± 198.91(10,326.00)	-	
	**G2**	12,541.31 ± 1238.21 (12,280.00)	14,484.07 ± 2607.69 (14,238.50)	***p*** **= 0.0022**
	**G3**	16,440.28 ± 1108.19 (16,198.00)	13,958.14 ± 1955.25 (13,686.50)	***p*** **= 0.0005**
	**G4**	12,673.83 ± 1706.30 (12,065.50)	-	
** *p* **		***p* < 0.0001** Differences between G1 vs. G2, G1 vs. G3, G1 vs. G4, G2 vs. G3, G3 vs. G4	*p* = 0.6098	


* Does not exhibit normal distribution; CRC = colorectal cancer; Se = sensibility; Sp = specificity; CEA= carcinoembryonic antigen; CA 19-9 = cancer antigen 19-9; PIVKA II = des-gamma-carboxy prothrombin; ucMGP = uncarboxylated matrix Gla protein; G1 = well-differentiated (low-grade) tumor; G2 = moderately differentiated (intermediate-grade) tumor; G3 = poorly differentiated (high-grade) tumor; G4 = undifferentiated (high-grade) tumor; the bold values are statistically significant.

**Table 4 ijms-25-04997-t004:** Predictive value of CEA, CA 19-9, PIVKA II, and ucMGP for colorectal cancer with or without metastasis.


**CEA, CA 19-9, PIVKA II, and ucMGP capacity to identify colorectal cancer—without taking into consideration metastasis presence**						
GLOBALLY	AUROC	95% Confidence Interval (lower-upper limit)	Cut off	Se	Sp	*p*
CEA	1	0.982–1.000	>3.9	100%	100%	**<0.001**
CA 19-9	0.976	0.945 to 0.993	>36.5	94.2%	100%	**0.0001**
PIVKA II	1	0.982 to 1.000	>42.9	100%	100%	**<0.001**
ucMGP	1	0.982 to 1.000	>3415	100%	100%	**<0.001**


**CEA, CA 19-9, PIVKA II, and ucMGP capacity to identify metastasis presence in colorectal cancer patients**						

	AUROC	95% Confidence Interval (lower-upper limit)	Cut off	Se	Sp	*p*
CEA	1.000	0.969 to 1.000	>157.3	100%	100%	**<0.001**
CA 19-9	1.000	0.969 to 1.000	>181.6	100%	100%	**<0.001**
PIVKA II	0.789	0.705 to 0.858	373.8	100%	65%	**0.0001**
ucMGP	0.608	0.515 to 0.696	>12,543	72.5%	48.7%	**0.0521**


AUROC = area under receiver operating characteristic curve; Se = sensibility; Sp = specificity; CEA= carcinoembryonic antigen; CA 19-9 = cancer antigen 19-9; PIVKA II = des-gamma-carboxy prothrombin; ucMGP = uncarboxylated matrix Gla protein; the bold values are statistically significant.

## Data Availability

The datasets referenced in this article are not accessible due to the informed consent agreement with the patients, wherein it is explicitly outlined that the data is intended solely for the purpose of this study and is not to be disseminated or utilized by third parties.

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
