# Peer review of "The Usefulness of Vitamin K-Dependent Proteins in the Diagnosis of Colorectal Carcinoma"

_ijms, 2024, doi:10.3390/ijms25094997_

Round 1

Reviewer 1 Report

Comments and Suggestions for Authors

The manuscript presents an observational case-control study aimed to compare standard tumor markers in colorectal cancer with some vitamin K-dependent proteins (ucMGP and PIVKA II).

In the Materials and Methods section only patients are presented with demographic and some clinicopathological data. The methodology of the study regarding the detection of vitamin K-dependent proteins, as well as of other paraclinical parameters is missing. The exact tests and reagents (catalogue number, company, etc. for detection of the proteins of interest should be precisely cited. Decision of Ethics Committee is not provided.

The Results are obtained from total of 200 patients, divided into three groups - without oncological pathology, with colorectal cancer without secondary liver metastasis, and colorectal cancer with secondary liver involvement. They are illustrated by 5 tables and show that ucMGP and PIVKA-II are significantly altered in patients with colorectal cancer especially in the early stages of the disease. They are studied in parallel with CEA, CA 19-9, tumor grade, and localization. In cancer patiens with no metastasis, significant inverse correlation is found between CEA and PIVKA II and ucMGP. The relationship is not proven in metastatic patients.

In the Discussion section the observation of lack of discrimination capacity between patients with cancer vs those without cancer regarding the classical tumor markers and PIVKA II & ucMGP could be further critically assessed. Another possible limitation of the study is the lack of data on microsatellite instability and mutation status that could also influence the results.

The references are correctly cited.

The manuscript is acceptable for publication after minor revisions.

Author Response

Dear Professor,

Please consider our revised manuscript, “The usefulness of vitamin K-dependent proteins in the diagnosis of colorectal carcinoma”.

First, we appreciate the time and effort that you have taken in our manuscript and the constructive criticism given. We are truly grateful for the insightful comments, as well as for the improvement suggestions, which we found quite helpful. We have addressed all the concerns.

We have incorporated all recommendations in our revised paper. Consequently, you will find both content and structural changes in the manuscript.

We hope that you will find the revised version of our paper considerably improved and complying with your suggestions, with a much better flow, coherence, and narrative.

See below, point-by-point, our responses. 

Thank you again.

Best regards,

Authors - Mirela-Georgiana Perné, Adela-Viviana Sitar-Tăut, Olga Hilda Orășan, Vasile Negrean, Călin Vasile Vlad, Teodora-Gabriela Alexescu, Mircea Vasile Milaciu, Lorena Ciumărnean, Răzvan Dan Togănel, Gabriel Emil Petre, Ioan Șimon,  Alexandra Crăciun

Comment: 

In the Materials and Methods section only patients are presented with demographic and some clinicopathological data. The methodology of the study regarding the detection of vitamin K-dependent proteins, as well as of other paraclinical parameters is missing. The exact tests and reagents (catalogue number, company, etc. for detection of the proteins of interest should be precisely cited. Decision of Ethics Committee is not provided.

Response:

Sample processing method and the manufacturer of the kits have been mentioned under the Materials and Method section. Ethical Commete decision date and approval number have been provided in the manuscript.

Comment: 

The Results are obtained from total of 200 patients, divided into three groups - without oncological pathology, with colorectal cancer without secondary liver metastasis, and colorectal cancer with secondary liver involvement. They are illustrated by 5 tables and show that ucMGP and PIVKA-II are significantly altered in patients with colorectal cancer especially in the early stages of the disease. They are studied in parallel with CEA, CA 19-9, tumor grade, and localization. In cancer patiens with no metastasis, significant inverse correlation is found between CEA and PIVKA II and ucMGP. The relationship is not proven in metastatic patients.

Response: 

Yes, it is true, that we have not presented the complete data about the correlations between CEA and PIVKA II and ucMGP in metastatic patients – because of the large table dimensions and the greater visual impact of the figures. If you consider it necessary, we can upload the table as supplementary material.

We consider that apparent paradox is very possible because in the metastatic phase also the liver function is affected, so the PIVKA II and ucMGP values will be altered; on the other side, the classical tumoral markers are known to increase in a parallel way with disease severity and metastasis appearance.

Comment: 

In the Discussion section the observation of lack of discrimination capacity between patients with cancer vs those without cancer regarding the classical tumor markers and PIVKA II & ucMGP could be further critically assessed. Another possible limitation of the study is the lack of data on microsatellite instability and mutation status that could also influence the results.

Response: 

Your suggestions are very pertinent. This study is just the first one, we are focusing on research in this area; for sure data regarding the mutation status are important in the evaluation of colorectal cancer patients. At the study beginning, the data about PIVKA II and ucMGP were so scarce, that we were not able to perform extensive research. However, this study seems to be a backbone for the next research. If you have some ideas, please do not hesitate to contact us for a possible collaboration.

Reviewer 2 Report

Comments and Suggestions for Authors

1. The introduction needs to be rewritten, it's too wordy.

2. Symbols are written in a standardized manner, e.g., p-values for significant differences need to be italicized。

3. Is it contradictory that the discussion section begins by stating that CEA and CA 19-9 tumor markers alone are not recommended for screening, followed by a reference to these two markers as currently represented?

4. The Discussion section mentions that only one article has been presented so far on the relationship between the presence of colorectal tumors with secondary dissemination in the liver and PIVKA-II serum levels. Our results are contrary to those reported by Kato, who reported a high value of PIVKA II in patients with colorectal cancer and metastases. Is PIVKA-II important?

Comments on the Quality of English Language

"Moderate editing is needed for the English."

Author Response

Dear Doctor,

Please consider our revised manuscript, “The usefulness of vitamin K-dependent proteins in the diagnosis of colorectal carcinoma”.

First, we appreciate the time and effort that you have taken in our manuscript and the constructive criticism given. We are truly grateful for the insightful comments, as well as for the improvement suggestions, which we found quite helpful. We have addressed all the concerns.

We have incorporated all recommendations in our revised paper. Consequently, you will find both content and structural changes in the manuscript.

We hope that you will find the revised version of our paper considerably improved and complying with your suggestions, with a much better flow, coherence, and narrative.

See below, point-by-point, our responses. 

Thank you again.

Best regards,

Authors - Mirela-Georgiana Perné, Adela-Viviana Sitar-Tăut, Olga Hilda Orășan, Vasile Negrean, Călin Vasile Vlad, Teodora-Gabriela Alexescu, Mircea Vasile Milaciu, Lorena Ciumărnean, Răzvan Dan Togănel, Gabriel Emil Petre, Ioan Șimon,  Alexandra Crăciun

Comment: 

The introduction needs to be rewritten, it's too wordy.

Response: 

The manuscript has been revised and the language improved. We have reduced the number of words to our best capabilities, however, as we have to introduce multiple subjects, and also to make sure that the readers have the best possible understanding, it is hard to further shorten the introduction section. We hope you would approve the section as it is. 

Comment: Symbols are written in a standardized manner, e.g., p-values for significant differences need to be italicized

Response: We have performed the adjustment as advised.

Comment: 

Is it contradictory that the discussion section begins by stating that CEA and CA 19-9 tumor markers alone are not recommended for screening, followed by a reference to these two markers as currently represented?

Response: 

What we intend to say is the fact that the use of CEA and CA 19-9 tumor markers ALONE is not recommended for screening, but their association with lower endoscopy and imaging can bring plus value. Our study aims to detect new markers that will be useful for screening, new blood markers that can be used separately.  Also, the published studies in the past demonstrated that CEA and CA 19-9 are useful tumor markers in the post-operative follow-up of patients with colorectal cancer.

Comment: 

The Discussion section mentions that only one article has been presented so far on the relationship between the presence of colorectal tumors with secondary dissemination in the liver and PIVKA-II serum levels. Our results are contrary to those reported by Kato, who reported a high value of PIVKA II in patients with colorectal cancer and metastases. Is PIVKA-II important?

Response: 

Most studies in the specialized literature have shown that the serum values ​​of PIVKA II are elevated in patients with hepatocarcinoma, but without being able to state that PIVKA II is a tumor marker superior to AFP (alpha-fetoprotein) in the detection of this neoplasia.

Considering that PIVKA II is an extrahepatic protein dependent on vitamin K that is synthesized in the liver, and serum values ​​can be influenced by other hepatopathies, not just hepatocarcinoma, by determining PIVKA II we wanted to observe if its serum values ​​are influenced by the presence of colorectal cancer with secondary liver determinations.

What we intend to say – about the Kato study- is the fact that in his study, in a case of CRC with metastasis, the PIVKA II value was increased; in our study, the PIVKA II presented a U-shape evolution. We need more studies to establish a clear relationship between PIVKA II – CRC - metastasis.

 Also considering the fact that it is an extrahepatic protein dependent on vitamin K, like MGP, yes, we think it is important, but further studies are needed to state this with certainty.

Round 2

Reviewer 2 Report

Comments and Suggestions for Authors

We thank the author for his careful revision of the manuscript. Minor editing required for English.

Comments on the Quality of English Language

Minor editing required for English.